# Spatiotemporal coupling and decoupling of gene transcription with DNA replication origins during embryogenesis in *C. elegans*

Ehsan Pourkarimi[1], James M Bellush[1,2], Iestyn Whitehouse[1]*

[1]Molecular Biology Program, Memorial Sloan Kettering Cancer Center, New York, United States; [2]BCMB Graduate Program, Weill Cornell Medical College, New York, United States

**Abstract** The primary task of developing embryos is genome replication, yet how DNA replication is integrated with the profound cellular changes that occur through development is largely unknown. Using an approach to map DNA replication at high resolution in *C. elegans*, we show that replication origins are marked with specific histone modifications that define gene enhancers. We demonstrate that the level of enhancer associated modifications scale with the efficiency at which the origin is utilized. By mapping replication origins at different developmental stages, we show that the positions and activity of origins is largely invariant through embryogenesis. Contrary to expectation, we find that replication origins are specified prior to the broad onset of zygotic transcription, yet when transcription initiates it does so in close proximity to the pre-defined replication origins. Transcription and DNA replication origins are correlated, but the association breaks down when embryonic cell division ceases. Collectively, our data indicate that replication origins are fundamental organizers and regulators of gene activity through embryonic development.

*For correspondence: whitehoi@
mskcc.org

**Competing interests:** The authors declare that no competing interests exist.

## Introduction

In many developing organisms, early rounds of DNA replication occur in the absence of zygotic gene transcription: in both *Drosophila* and *Xenopus* embryos, the rapid genome duplication in early cycles is facilitated by the use of many, closely spaced, replication initiation sites (*Blumenthal et al., 1974*; *Callan, 1974*). Within *Drosophila* and *Xenopus*, the onset of zygotic transcription is accompanied by a marked increase in S phase length, caused by a reduction in the number of replication initiation events, rather than changes in the rate of replication fork progression (*Blumenthal et al., 1974*). Numerous studies in metazoa have found particular chromatin signatures that typify transcriptionally active chromatin are enriched at replication initiation sites and also influence the relative time in S phase that origins are active (*MacAlpine and Almouzni, 2013*). Yet a specific signature within chromatin that marks distinct sites in a genome where replication initiates has not been identified (*Urban et al., 2015*).

We chose to investigate DNA replication within developing *C. elegans* embryos, with the goal of characterizing signatures of replication origins and understanding replication patterns through early development. Our approach relies on transient depletion of DNA ligase I, which results in the accumulation of Okazaki fragments on the lagging strand (*Smith and Whitehouse, 2012*; *McGuffee et al., 2013*). By sequencing Okazaki fragments we, and others, have provided high-resolution maps of DNA replication across budding yeast (*McGuffee et al., 2013*) and human genomes

(*Petryk et al., 2016*). One advantage of sequencing Okazaki fragments is that replication signatures can be deduced from a population of cells growing at steady state. Thus, providing that Okazaki fragments can be harvested in sufficient quantity, DNA replication can be mapped in cells, tissues or organisms without need for synchronization (*Smith and Whitehouse, 2012*).

## Results and discussion

Okazaki fragments were purified from mixed early embryos (ME) following RNAi depletion of *lig-1*, which is the sole *C. elegans* DNA ligase responsible for ligating Okazaki fragments. To minimize genome instability and prevent embryonic lethality associated with complete *lig-1* depletion, we determined an optimal level of depletion to maintain normal development, while enriching for fragments by dilution of the *lig-1* RNAi bacteria (*Figure 1A*, *Figure 1—figure supplement 1*). Resolution of labeled DNA harvested from *lig-1* depleted embryos on a denaturing agarose gel, revealed a highly periodic pattern of Okazaki fragment size, similar to that of the nucleosome repeat uncovered by MNase digestion (*Figure 1A*). While the periodicity is far more pronounced in *C. elegans*, the size distribution of Okazaki fragments matches those observed in budding yeast, indicating that the coupling of lagging strand synthesis and nucleosome assembly is likely conserved (*Smith and Whitehouse, 2012*; *Yadav and Whitehouse, 2016*).

Next, we analyzed the Okazaki fragments by deep sequencing using a protocol that preserves strand identity (*Smith and Whitehouse, 2012*). Mapping to the worm WS220 reference genome, we find a clear strand bias, which matches the expected distribution of Okazaki fragments synthesized on the lagging strand: replication origins are at sites of transition from Watson to Crick strand reads (*Figure 1B*) (*McGuffee et al., 2013*). While the data is noisier than budding yeast – in part due to the high A:T content of the genome, residual DNA ligase activity and the complexity of the sample preparation – we identified >2000 replication origins using a modified mapping protocol (*McGuffee et al., 2013*). Median spacing between origins is 40 kb, and 96% of origins are within 100 kb of another origin; the median efficiency – that is, the likelihood that a given origin is utilized during S phase, is ~50%, although we note that numerous origins are fired in most cells in the population (*Figure 1—figure supplement 2A and B*). In contrast to budding yeast, in which initiation is confined to a narrow range (*McGuffee et al., 2013*), replication in *C. elegans* apparently initiates within broad zones at most origins, similar to human cells (*Figure 1—figure supplement 3*) (*Petryk et al., 2016*; *Dijkwel et al., 1991*). Replication origins potentially influence gene organization: genes near origins tend to be shorter and genes flanking origins show a pronounced bias to ensure that replication fork progression and transcription are co-directional (*Figure 1—figure supplement 4A and B*) (*Petryk et al., 2016*).

Having generated a high-resolution map of replication origins, we next considered whether particular chromatin signatures are enriched near origins. Thus, we analyzed the abundance of several histone modifications that were mapped in embryos at similar stages (*Gerstein et al., 2010*). As *Figure 2* shows, there is a remarkable concordance between the location of replication origins and acetylation of histone H3 and methylation of H3 at lysine 4 (K4). Analyzing H3 K27ac (H3 K4me2 is near identical), we find that 75% of efficient origins (>30% efficiency) are within 1 kb of the histone mark and most H3 K27ac peaks overlap origins (*Figure 2B*). Further, the most efficient replication origins are associated with sites with the most H3K27 acetylation (*Figure 2C*, *Figure 2—figure supplement 1*). By surveying the abundance of various modifications present at replication origins, we tested whether other modifications behaved similarly to H3K27ac (*Figure 2D*); this revealed that acetylation of H3 at K18, K23, K27 and dimethylation of H3 at K4 are not only present at origins, but the level of modification increases according to origin efficiency. Significantly the histone modifications, we find most strongly associated with origins are those that define gene enhancers in metazoan genomes (*Ho et al., 2014*).

Given that ME embryos are transcriptionally active, our finding that replication initiates near histone marks linked with enhancers indicates that origins are localized to 'accessible' transcriptionally active regions. In this scenario, replication origins become defined once zygotic transcription has initiated; DNA replication in pre-gastrula embryos (PG), where transcription is limited and non essential (*Robertson and Lin, 2015*), would presumably initiate from many, seemingly random sites – much like the apparent situation in *Drosophila* and *Xenopus* embryos prior to the Mid Blastula Transition (MBT). To investigate this, we harvested pre-gastrula embryos from a hyperactive egg laying strain

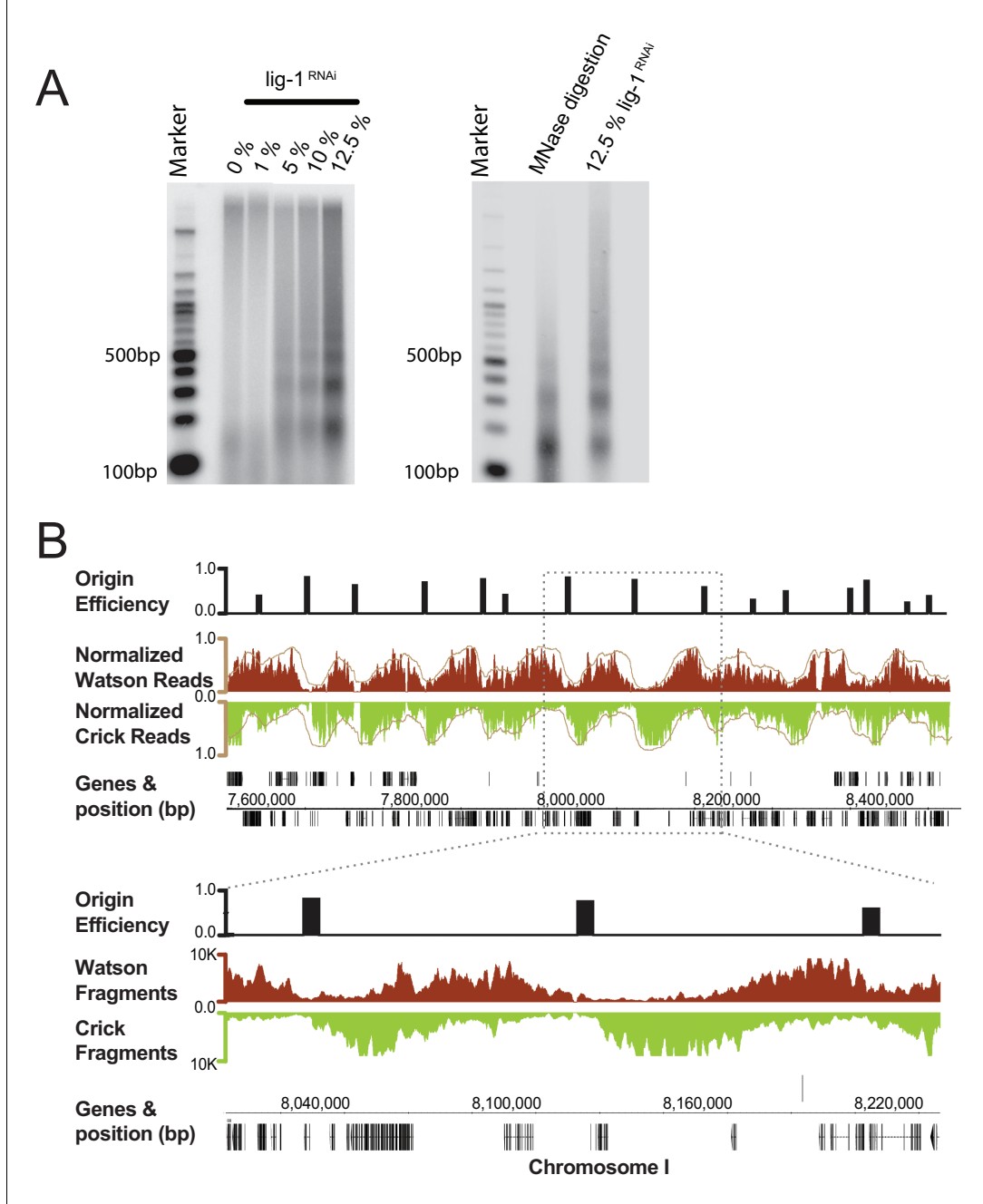

**Figure 1. Purification and sequencing of Okazaki fragments from C. elegans.** (A) Left panel: increasing DNA ligase I (*lig-1*) depletion by RNAi results in the generation of small DNA molecules with periodic size distribution. Right panel: comparison of the putative Okazaki fragments with DNA resultant from a Micrococcal Nuclease digestion of chromatin, reveals both species have similar periodicity. (B) Fragments mapped to the *C. elegans* genome display a characteristic strand bias expected of Okazaki fragments. Regions of high Watson-strand signal (red) that transition to regions of high Crick-strand signal (green) are replication origins; normalized signal is shown in brown. Mapped replication origins are black bars above the Okazaki fragment traces and are shown as 5 kb blocks centered on the origin midpoint to aid visualization. Annotated genes and their coordinates are shown below the fragment traces.

The following figure supplements are available for figure 1:

**Figure supplement 1.** Average progeny number in *C. elegans* treated with 100% and 10% lig-1 RNAi.

**Figure supplement 2.** Global measures of origin efficiency and spacing.

*Figure 1 Purification and sequencing of Okazaki fragments from C elegans continued on next page*

*Figure 1 Purification and sequencing of Okazaki fragments from C elegans continued*

**Figure supplement 3.** Heatmap illustrating the varying size of the 'initiation zone' at replication origins.
**Figure supplement 4.** Gene orientation around replication origins.

(PG: median 9, mean 13 cells/embryo, n = 50; *Figure 3A*) and late embryos from wild-type worms (L: ~200–558 cells/embryo), which represent different levels and extent of gene expression (*Figure 3B*). We compared origin location and efficiency for each population and found that the patterns of replication were highly similar (*Figure 3C*) – especially for the most efficient origins (*Figure 3D*). Evidently, even though gene transcription is fundamentally altered through this time course, replication origin usage is globally similar through early development. Thus, the establishment of specific replication origins in *C. elegans* occurs prior to the broad onset of zygotic transcription – meaning that the replication origins we define are unlikely to be dependent upon transcription. This finding is in keeping with data showing that isolated embryonic cells treated with α-amanitin divide normally up to approximately 100 cells (*Edgar et al., 1994*). Indeed, S phase length, lineage-specific cell cycle timing, asymmetric cell division and cleavage patterns all proceed normally in the absence of transcription in pre-gastrula early embryos (*Robertson and Lin, 2015*; *Edgar and McGhee, 1988*; *Edgar et al., 1994*).

Coupling of DNA replication with gene enhancers within rapidly dividing embryonic cells likely imposes particular constraints upon genome duplication. S phase in pre-gastrula embryonic cells ranges from ~10 to 25 min, depending on the lineage (*Edgar and McGhee, 1988*). In an ideal system, where all origins fire in each S phase (100% efficiency), complete genome duplication could be achieved in ~15 min if origins are uniformly spaced every ~75 kb and replication forks progress at 2.5 kb/min. We find that the median origin spacing is ~40 kb (*Figure 1—figure supplement 1*), which reflects that not all origins are fired within each S phase. Given that DNA replication appears to initiate at enhancers, the requirement for closely spaced origins to achieve timely genome duplication would seemingly constrain the positions of enhancers across the genome. Thus, enhancers, and the genes they regulate, may be organized to facilitate the execution of the DNA replication and transcription programs in rapidly dividing embryonic cells. To investigate this, we first performed a gene ontology analysis of genes at varying distances from replication origins: this revealed that genes whose products are involved growth, embryonic development, cell cycle, gene expression, and chromatin are clustered near replication origins (*Figure 4A* and *Figure 4—figure supplement 1*). Next, we compared origins with the whole embryo transcriptome time series derived from individual embryos from the one-cell stage through hatching (*Hashimshony et al., 2015*). For each of the 50 time points, we removed maternally derived transcripts and plotted the sum of normalized transcript levels for each gene relative to the midpoint of replication origins. As shown in *Figure 4B* and *Figure 4—figure supplement 2*, transcript abundance accumulates through the early time points with no apparent trends in the data; but, beginning at time point ~9 and coincident with gastrulation, there is an apparent clustering of actively transcribed genes near replication origins. The association of transcription with origins spans to time point ~25 and broadly scales according with origin efficiency (*Figure 4—figure supplement 2*). Beyond time point 25, the association breaks down; yet, as gene expression changes, the pattern evolves such that from time point 40 onwards, sites of active transcription now appear anti-correlated with replication origins. As we have shown in *Figure 3*, replication origins are defined prior to the onset of global zygotic gene activation, yet *Figure 4B* reveals that when the genome becomes broadly transcriptionally active, transcription generally occurs in close proximity to the pre-defined origins. At time point 25, which is ~300 min into embryogenesis, transcription begins to shift away from the origins, and by 500 min, peak transcription occurs some 15–25 kb from the origin. Since the median spacing between origins is 40 kb, the shift in transcription late in embryogenesis ensures that gene transcription is as far removed as possible from sites of replication initiation. The alteration in transcription that occurs after ~300 min is coincident with ventral enclosure and has been shown previously to underlie a developmental milestone wherein the 'morphogenesis' transcriptome is activated (*Levin et al., 2012*). Given the close association between replication origins and expressed genes, we considered whether global alterations in the transcriptome may be related to origin activity. Thus, we compared the origin/transcript

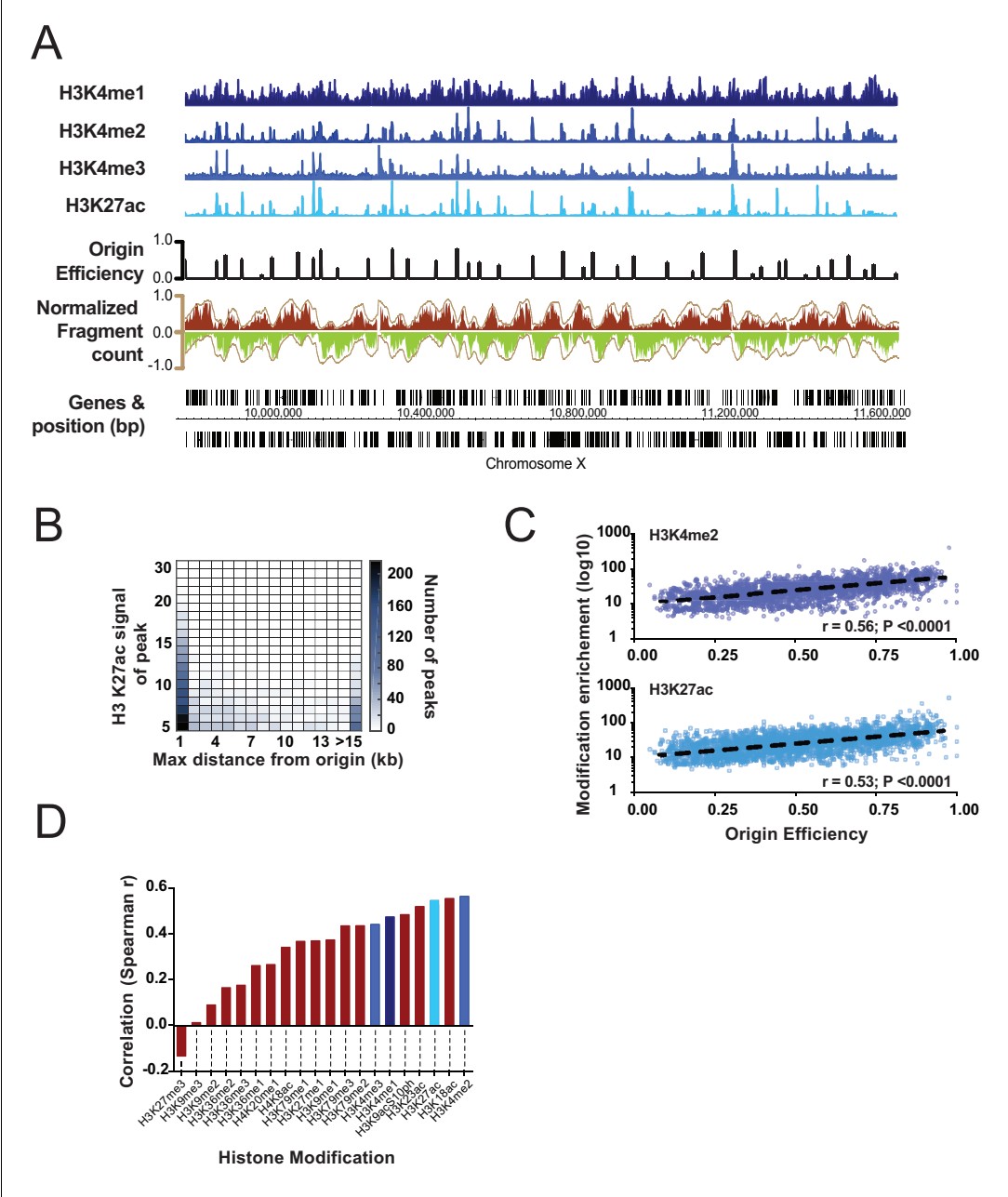

**Figure 2.** Correlation of select histone modifications with replication origins. (**A**) Representative ChIP signal for select histone modifications are shown for a ~1 Mb region of Chromosome X. Mapped replication origins (black) are broadly coincident with peaks in the ChIP signal. Okazaki fragment reads are shown below and are colored according to *Figure 1B*. (**B**) Correlation between H3K27ac and replication origins is displayed as a heatmap. Replication origins and H3K27ac peaks were defined computationally and the distance between peaks measured using Intervalstats (Materials and ethods); the number of H3K27ac peaks that lie within a maximum distance, defined on the x-axis, is plotted as heatmap; the relative intensity (fold enrichment over background) of the ChIP signal at H3K27ac peaks is plotted on the y-axis. (**C**) ChIP signal scales with increasing origin efficiency; level of histone modifications at replication origins (±2.5 Kb) is plotted for H3K27ac and H3K4me2 relative to the efficiency of the origin. (**D**) Data were analyzed as **C** to compute the correlation of several histone modifications with replication origin efficiency; blue shades correspond to histone marks shown in **A**.

The following figure supplement is available for figure 2:

**Figure supplement 1.** Heatmap illustrating the relationship between H3K27ac and replication origins.

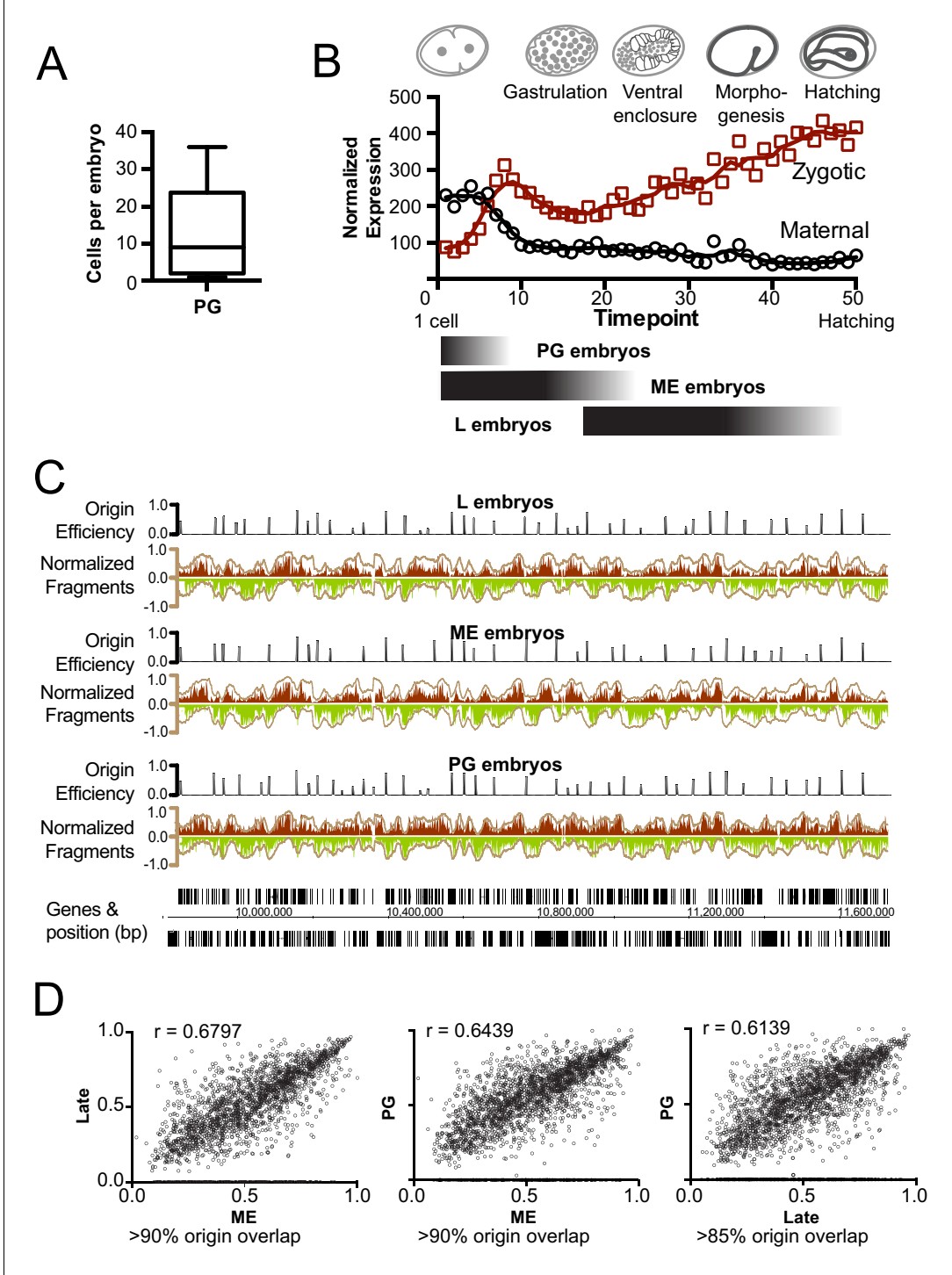

**Figure 3.** Mapping replication origins through early embryogenesis. (**A**) Box plot showing the number of cells/embryo for a representative sample of pre-gastrula (PG) embryos, whiskers indicate the range of values. (**B**) Graph showing maternal and zygotic transcripts through embryogenesis. Data from whole embryo transcriptome from Hashimshony et al (*Hashimshony et al., 2015*) were normalized so that the sum expression for each gene through the time course = 1; total transcript levels for all genes at time point is shown. Below, black shaded bars illustrate the approximate age range for the purified embryo populations; late (L), mixed early (ME) and pre-gastrula (PG). (**C**) Okazaki fragment sequencing of DNA from L, ME and PG embryos revels broadly similar replication patterns. Graphs colored according *Figure 1B*. (**D**) Scatterplot comparing origin the efficiencies of overlapping origins within the L, ME and PG datasets. Spearman correlation (r) is shown on graph. The % of origins that overlap spatially is shown below.

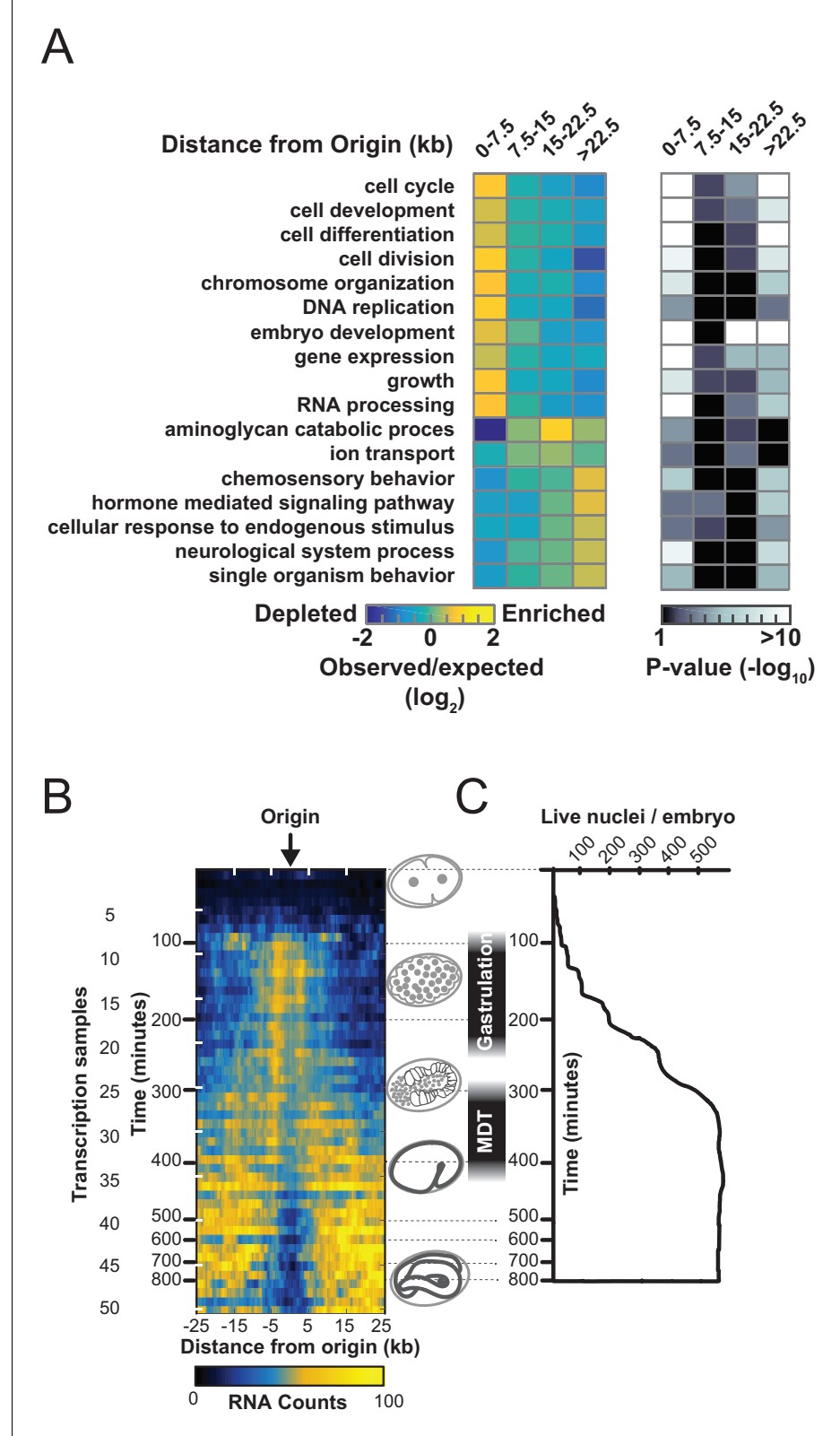

**Figure 4.** Gene transcription is coupled with replication origin use. (A) Gene ontology analysis for genes at varying distances from the midpoint of a replication origin. Gene ontologies were calculated using the Gene Ontology Consortium (http://geneontology.org/). Select GO terms with greatest significance are shown that lie within 'Biological Process' annotation datasets. Left, observed/expected log₂ ratio is shown at varying distances from
*Figure 4 continued on next page*

*Figure 4 continued*

replication origins for each ontology term as a heatmap: color key is below – yellow indicates enrichment, blue is depletion. Right, heatmap showing the calculated p-values (hypergeometric distribution) for the data on left; color key is below. (B) Normalized transcript abundance within ±25 kb of the replication origin was summed for the top 1000 origins for each of the 50 time points through embryogenesis. Data are displayed as a heatmap and arranged according to sample number shown on left, corresponding time in embryogenesis is also indicated. Timing of gastrulation and mid development transition (MDT) calculated by (*Levin et al., 2016*) are also shown. (C) Live cell nuclei per embryo as calculated by (*Sulston et al., 1983*) is plotted as a function of time through embryogenesis and aligned with **B**.

The following figure supplements are available for figure 4:

**Figure supplement 1.** Gene ontology analysis for genes at varying distances from the midpoint of a replication origin.

**Figure supplement 2.** Heatmap showing the relationship between transcribed genes and replication origins through embryogenesis.

profiles with a time course that measured living nuclei per embryo during embryogenesis from Sulston et al (*Sulston et al., 1983*) (*Figure 4C*). Significantly, the shift of transcription from origins that begins at ~300 min is coincident with the completion of the last wave of cell division that occurs within the embryo – indicating that the inactivation of replication origins may be coupled with the restructuring of the transcriptome to facilitate post replicative development.

Our finding that replication initiates at specific sites in PG embryos is in marked contrast to the seemingly random patterns observed in the rapid and synchronous cycles in both *Drosophila* and *Xenopus* embryos (*Blumenthal et al., 1974*; *Callan, 1974*). However, early development in *C. elegans* differs from these organisms: initial cell divisions are asynchronous, and the cell cycle does not slow abruptly near gastrulation (*Bao et al., 2008*). Thus, the rapid establishment of the defined replication landscape in *C. elegans* may be related to the very early stage at which cell lineages are specified. Indeed, several histone modifications, including H3K27ac and H3K4me2, are potentially passaged with the maternal genome and are known to be present in two cell embryos (*Arico et al., 2011*; *Samson et al., 2014*). Replication origins appear to be tied with gene expression: in general terms, origin proximal genes experience the first wave of zygotic transcription and are expressed in replicating cells; origin distal regions encode genes whose transcription is delayed until later in development when DNA replication has ceased. Our results indicate that the *C. elegans* genome has evolved to couple embryonic transcription with replication; yet, such coupling would appear to impose constraints on a system that needs not only to transcribe a diverse array of genes, but also replicate the ~100 mb genome in ~15 min. *C. elegans* seemingly resolves this issue as follows: first, replication origins appear to be co-localized with gene enhancers which can activate transcription of genes at a distance; second, genes required for cell division and early development are clustered near replication origins (*Figure 4—figure supplement 1*). Such association is likely to be functionally important in rapidly cycling cells with limited gap phases: most obviously, genes near to origins such as histones, will be replicated at the start of S phase and have a higher effective copy number than those further away. Relatedly, the passage of the replication fork may trigger genes to be reactivated that were presumably repressed during mitosis; thus ensuring replication-transcription conflicts do not occur. Finally, the inactivation of replication origins may serve as a trigger to relieve the gene expression program from replication origin-associated spatial constraints – allowing the expression of genes whose products are required for the many aspects of development that occur in the absence of cell division. The spatiotemporal coupling and uncoupling of gene expression with replication may be a general principal underlying the profound transcriptional and morphological changes that occur during embryogenesis across species: genes expressed early are characterized by proteins involved in proliferation, while those expressed late are typically involved in differentiation (*Levin et al., 2016*). Separating early and late genes is a comparatively short phase termed the 'mid-developmental transition' (*Levin et al., 2016*); in *C. elegans,* we find that this transition is coincident with the uncoupling of transcription from replication origins (*Figure 4B,C*). It will be of great

interest to learn whether this is a ubiquitous feature of the mid-developmental transition across species.

## Materials and methods

### *C. elegans* maintenance and strains

All *C. elegans* strains were maintained at 20°C on NGM plates with OP50 *E. coli* strain, as previously described (*Brenner, 1974*). Strains used in this study:

The wild-type strain corresponds to Bristol N2 (RRID:CGC_N2), CG21 *egl-30(tg26) I; him-5 (e1490) V* (RRID:CGC_CG21).

### Knocking down lig-1 using RNAi

We used lig-1 RNAi generated by the Ahringer laboratory and used a modified version of RNAi feeding protocol (*Fraser et al., 2000*). *lig-1* and control (empty vector) RNAi-expressing bacteria were grown overnight at 37°C in LB medium supplemented with 50 µg/ml Ampicillin and 10 µg/ml tetracycline. Bacterial cultures were transferred to fresh LB medium supplemented with 50 µg/ml Ampicillin and were grown at 37°C until reaching OD 1.0. RNAi expression was induced with 1 mM IPTG for 1 hr at room temperature.

To attenuate *lig-1* associated genome instability and prevent sterility associated with complete *lig-1* depletion, *lig-1* RNAi expressing bacteria was diluted 1:10 with control (empty vector) bacteria to a final volume of 500 ml. Diluted bacteria were harvested by centrifugation and added to *C. elegans* S-Basal liquid media containing 150,000–200,000 well synchronized early L3 staged N2 worms (see below). N2 worms were grown in S-Basal liquid culture containing diluted *lig-1* RNAi at 20°C until adulthood. Pre Gastrula embryos were harvested from early L3 staged egl-30 mutants that were grown on 1 mM IPTG plates as described previously (*Fraser et al., 2000*). Gravid adults were harvested and their embryos were collected by bleaching. Additional *lig-1* dilution ratios (*Figure 1a*) were generated as above.

To collect synchronous L1 populations, 10,000 worms were grown on NGM plates. Their progeny were washed with M9 buffer and passed through 11 µM nylon net filters (Millipore Ltd. NY1104700). Approximately 150,000 synchronized L1 stage worms were grown in liquid media supplemented with OP50 bacteria until reaching L3 stage. Synchronized L3 staged worms were collected by centrifugation, followed with three times washing with M9 buffer containing 50 µg/ml Ampicillin. For progeny counting L3 staged worms where grown on 1 mM IPTG plates.

### Genomic DNA purification and Okazaki fragment labeling

Okazaki fragments were purified as previously described with slight modifications (*Smith and Whitehouse, 2012*). Samples were resuspended in 480 µl Lysis buffer (50 mM Tris-HCL, pH 8.0, 50 mM EDTA, 100 mM NaCl, 1.5% Sarkosyl, 1% SDS) and incubated with 200 µg proteinase K at 42 degree overnight. Digested proteins and peptides were precipitated by addition of 200 µl 5 M KOAc and centrifugation at 16,000 *g* for 30 min at 4°C. Genomic DNA was precipitated by adding 500 µl iso-propanol and spinning at 16,000 *g* at 4°C for 10 min. Genomic DNA pellets were washed with 70% ethanol and then resuspended in 300 µl STE (10 mM Tris-HCL, PH8.0, 1 mM EDTA, 100 mM NaCl). Residual RNA was digested by addition of 5 U RiboShredder RNase Blend (Epicentre) at 37°C overnight. Genomic DNA was precipitated with 30 µl 3 M NaOAc and 1.7 ml ethanol pelleted at 10,000 *g* for 10 min at 4°C. DNA pellets were washed with 70% ethanol and resuspended in 30 µl TE (10 mM Tris-HCl pH 7.5, 0.1 mM EDTA) and stored over night at 4°C to allow complete resuspension of genomic DNA. Samples were stored at −80°C. Radiolabeling of Okazaki fragments and denaturing gel electrophoresis were followed as previously described (*Smith and Whitehouse, 2012*). Input genomic DNA for labeling was 2–3 µg.

### Okazaki fragment purification

Okazaki fragments were purified from genomic DNA of Ligase I depleted *C. elegans* embryos, by ion exchange chromatography, similar to the procedure previously described (*Smith and Whitehouse, 2012*). *C. elegans* Okazaki fragments were enriched in the 750–850 mM NaCl fractions. These fractions were pooled and DNA precipitated for sequencing library preparation.

## Sequencing library generation

Three hundred nanograms purified Okazaki fragments were used to generate sequencing libraries similar to the previously optimized protocol (*Smith and Whitehouse, 2012*). Following fragment ligation the total reaction was loaded on a 2% agarose gel; fragments corresponding to ~200–700 bp were purified from the gel using QiAquick gel extraction kit (Qiagen). Purified ligated fragments were amplified by (PCR 16) cycles using custom Illumina or Ion Torrent sequencing oligos.

## MNase digestion of *C. elegans* embryos

To isolate nucleosomal DNA through in vivo micrococcal nuclease digestion, embryos were harvested from gravid N2 adults by bleaching. Embryos were permeabilized by digesting with chitinase (Sigma, St. Louis, MO; C6137) followed by pronase (EMD Millipore, 537088-50KU) each for 3 min at 37°C (*Edgar and Goldstein, 2012*). Embryos were then pelleted and washed 3X with Egg Buffer (118 mM NaCl, 48 mM KCl, 3 mM $CaCl_2$, 3 mM $MgCl_2$, 5 mM HEPES, pH 7.3). Embryos were fixed with a 1% formaldehyde solution for 30 min and were quenched by adding glycine to a 125 mM final concentration and washed with NP-S buffer (0.075% NP-40, 50 mM NaCl, 10 mM Tris pH 7.4, 5 mM $MgCl_2$, 1 mM $CaCl_2$, 1 mM $\beta$-mercapatoethanol) for MNase digestion. Thirty-five thousand fixed embryos were digested with 100 U of MNase (Worthington, LS004798) for 3 min at 23°C. After the reaction was terminated, the samples were treated with RNase A and proteinase K before incubation at 65°C for 12 hr to reverse crosslinks. Samples were phenol extracted and the DNA subsequently precipitated (*Ooi et al., 2010*). DNA was visualized on 1.3% denaturing agarose gel following labeling with T4 PNK (NEB) and radiolabeled γ-ATP.

## Genomics protocols

Sequencing was performed using Ion Torrent (Proton) or Illumina (HiSeq) platforms. At least one biological replicate was performed for each DNA replication map; Ion Torrent (Proton) generally had a higher background and lower reproducibility than Illumina so direct comparisons between different embryonic stages were made using only the Illumina data (*Figure 3*). Sequencing reads were mapped to the WS220 genome using Bowtie2 with –local function (RRID:SCR_005476). Reads with q < 30 were removed using Samtools (RRID:SCR_002105). Remaining reads were binned in 100 bp intervals using Bedtools (RRID:SCR_006646), maintaining strand identity. Data were partially smoothed by calculating the median with a sliding window of 1.5 kb (*Royce et al., 2007*). Data were normalized (such that the sum of Watson and Crick reads = 1) and positions of origins were mapped using custom program described earlier, using a 12 kb window (*McGuffee et al., 2013*). Because replication initiates in a broad zone at most origins, we found that the method of McGuffee significantly underestimated the true origin efficiency; therefore, origin efficiencies are defined by the maximum value of normalized reads within 20 kb of the origin. The distance between normalized maxima on the Watson and Crick strands defined the size of transition zone at origins. Histone modification data used in *Figure 2* were downloaded from ModEncode consortium (http://www.modencode.org/). Regions of ChIP enrichment were calculated using Macs2 (*Zhang et al., 2008*) with default parameters. Regions of enrichment were defined as peaks with >5 fold enrichment over background. Associations of ChIP peaks with replication origins was calculated using intervalstats (*Chikina and Troyanskaya, 2012*); replication origins were defined as a 5 kb region centered on the origin midpoint. Transcriptomics data from the whole embryo time course from (*Hashimshony et al., 2015*) were used in this study. Data for each gene were individually normalized such that the sum of all 50 time points = 1. Data were then clustered (unsupervised) and genes with highest transcription within the first five time points were defined as 'maternal' all other expressed genes are defined as 'zygotic'.

## Accession number

All sequencing data are availed at Gene Expression Omnibus with under accession number GSE90939.

## Acknowledgements

We thank D Remus, K Marians, members of the Whitehouse lab at MSKCC; Anton Gartner (University of Dundee), David MacAlpine (Duke), Duncan Smith (NYU) for comments on the manuscript and Zhirong Bao (MSKCC) for providing advice and reagents. We are thankful to the Caenorhabditis Genetics Center (CGC) for supplying strains and reagents. This work is supported by a NIH grant R01 GM102253 and an American Cencer Society Research Scholar Grant to IW and P30CA008748 awarded to MSKCC.

## Additional information

### Funding

| Funder | Grant reference number | Author |
|---|---|---|
| National Institutes of Health | GM102253 | Iestyn Whitehouse |
| National Institutes of Health | P30CA008748 | Iestyn Whitehouse |
| American Cancer Society | 128073-RSG-15-041-01-DMC | Iestyn Whitehouse |

The funders had no role in study design, data collection and interpretation, or the decision to submit the work for publication.

### Author contributions

EP, IW, Conceptualization, Resources, Data curation, Software, Formal analysis, Supervision, Funding acquisition, Validation, Investigation, Visualization, Methodology, Writing—original draft, Project administration, Writing—review and editing; JMB, Conceptualization, Data curation, Formal analysis, Validation, Methodology, Writing—review and editing

### Author ORCIDs

Ehsan Pourkarimi, http://orcid.org/0000-0001-9598-3465
Iestyn Whitehouse, http://orcid.org/0000-0003-0385-3116

## Additional files

### Supplementary files

• Supplementary file 1. All origin efficiencies: Origin efficiencies and locations are included in *supplementary file 1*.

### Major datasets

The following dataset was generated:

| Author(s) | Year | Dataset title | Dataset URL | Database, license, and accessibility information |
|---|---|---|---|---|
| Ehsan Pourkarimi, James M Bellush, Iestyn Whitehouse | 2016 | Replication maps of developing C. elegans embryos. | https://www.ncbi.nlm.nih.gov/geo/query/acc.cgi?acc=GSE90939 | Publicly available at the NCBI Gene Expression Omnibus (accession no. GSE90939) |

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
