## [Decision Letter]

Thank you for submitting your article "Spatiotemporal coupling and decoupling of transcription with DNA replication origins during embryogenesis in *C. elegans*" for consideration by *eLife*. Your article has been favorably evaluated by Jessica Tyler (Senior editor) and three reviewers, one of whom, Michael R Botchan (Reviewer 1) is a member of our Board of Reviewing Editors.

The reviewers have discussed the reviews with one another and the Reviewing Editor has drafted this decision to help you prepare a revised submission.

The Whitehouse lab had previously developed a strategy for mapping eukaryotic DNA replication origins based on the position of Okazaki fragment strand switching, which they used to successfully map replication origins as well as replication termination sites in budding yeast. In this paper, they further develop this technique to map origins in C.elegans. This is an important step forward, and, together with a paper from the Hyrien lab earlier this year using a related approach, represents the first time this technique has been applied to a metazoan. This has great value because this technique has been shown to accurately and sensitively identify origins in budding yeast. For this reason alone the reviewers believe this a significant paper. The authors go on to show that most of the origins used before gastrula are also used later in development. This is in contrast to the well-studied situation in *Xenopus* and *Drosophila* development where initiation sites before the mid-blastula transition are dense and poorly defined, perhaps truly random (although random is perhaps a poorly defined concept in this context), and then becomes restricted to specific sites only thereafter. They show that origins co-localise with the potential enhancers of genes expressed through the proliferative stages of development. They show that the position of these replication origins correlates with specific histone marks, and that these origins are active before significant levels of transcription are seen. Finally, they show that the cessation of transcription potentially associated with these cis acting enhancers correlates with the cessation of proliferation. The reviewers all felt that certain key points needed to be addressed before publication and these are listed.

1) The sites defined as "enhancers" seem to be assigned simply by the histone modification patterns. The reviewers are of course aware of the literature in higher eukaryotes that certain combinations of histone marks are predictive of enhancers but not for the worm. For example: it is not proven that sites of H3K27ac in worms are actually marking enhancers. The authors should either show that some of these sites are indeed transcription enhancers by mutation and effect transcription or nuance their discussion accordingly. If genome wide enhancers have been mapped perhaps they can be identified in a figure. In a related issue is #2.

2) The separation between transcription and replication regulation is important but perhaps difficult to prove at this point. Proteins that are binding to the "enhancer sites" may be critical for both functions and mutation of these cis sites could have effects on both processes. If the authors want to establish this as a strong conclusion perhaps they should use inhibitors of transcription (or approaches such as SiRNA to knock-down protein levels for factors essential for that process) and show that early in development replication patterns do not change. Otherwise the discussion should be modified.

3) Have the authors investigated potential phenotypes for ligase depletion at various times in development- such as fecundity or fertility?

---

## [Author Response]

[…]

*1) The sites defined as "enhancers" seem to be assigned simply by the histone modification patterns. The reviewers are of course aware of the literature in higher eukaryotes that certain combinations of histone marks are predictive of enhancers but not for the worm. For example: it is not proven that sites of H3K27ac in worms are actually marking enhancers. The authors should either show that some of these sites are indeed transcription enhancers by mutation and effect transcription or nuance their discussion accordingly. If genome wide enhancers have been mapped perhaps they can be identified in a figure. In a related issue is #2.*

The reviewers raise an important point and we agree that use of the term enhancers is not directly supported by empirical evidence from studies in *C. elegans*. To our knowledge there is no published study to systematically and functionally define enhancers in *C. elegans*. Our definition is based on a large body of data that clearly links enhancers with select histone modifications in other eukaryotes, but we agree with the reviewers comment and have rephrased our discussion accordingly. More descriptive terms such as “enhancer associated chromatin modifications” are now used throughout the manuscript.

*2) The separation between transcription and replication regulation is important but perhaps difficult to prove at this point. Proteins that are binding to the "enhancer sites" may be critical for both functions and mutation of these cis sites could have effects on both processes. If the authors want to establish this as a strong conclusion perhaps they should use inhibitors of transcription (or approaches such as SiRNA to knock-down protein levels for factors essential for that process) and show that early in development replication patterns do not change. Otherwise the discussion should be modified.*

This is a very important point, we agree that it is necessary to functionally separate transcription and replication in early developing embryos. Zygotic activation in *C. elegans* has been studied intensively and experiments similar to those suggested by the reviewers have been done previously. Importantly, inhibiting transcription with α-amanitin in isolated early embryonic cells was found not to affect timing of cell division, the asymmetric pattern of early divisions, or the pattern of cleavage. Lois G Edgar, Nurit Wolf and William B. Wood, Development 120, 443-451 (1994). To our knowledge, all previous studies indicate that zygotic transcription is inessential in early rounds of division. Due to technical limitations we cannot inhibit transcription by RNAi against Pol II as it will lead to complete sterility. Upon the referee’s suggestion we have emphasized these points in the text.

*3) Have the authors investigated potential phenotypes for ligase depletion at various times in development- such as fecundity or fertility?*

Knocking down *LIG-1* using 100% RNAi leads to rapid and complete sterility, therefore, we have optimized partial depletion of *lig-1* to achieve maximal viability whilst also enriching for fragments. Knocking down *lig-1* with 10% RNAi results in a loss of 30% of viable progeny (i.e. number of hatched embryos). As now shown as Figure 1—figure supplement 1.